# Exploring the Role of Rehabilitation Medicine within an Inclusion Health Context: Examining a Population at Risk from Homelessness and Brain Injury in Edinburgh

**DOI:** 10.3390/ijerph21060769

**Published:** 2024-06-14

**Authors:** Edwin Eshun, Orla Burke, Florence Do, Angus Maciver, Anushka Mathur, Cassie Mayne, Aashik Ahamed Mohamed Jemseed, Levente Novak, Anna Siddique, Eve Smith, David Tapia-Stocker, Alasdair FitzGerald

**Affiliations:** 1Department of Rehabilitation Medicine, Astley Ainslie Hospital, NHS Lothian, 133 Grange Loan, Edinburgh EH9 2HL, UK; alasdair.fitzgerald@nhs.scot; 2Edinburgh Medical School, College of Medicine & Veterinary Medicine, University of Edinburgh, 47 Little France Crescent, Edinburgh EH16 4TJ, UKa.a.mohamed-jemseed@sms.ed.ac.uk (A.A.M.J.);

**Keywords:** rehabilitation, traumatic brain injury, inclusion health, homelessness, multimorbidity

## Abstract

People experiencing homelessness are at risk from a number of comorbidities, including traumatic brain injury, mental health disorders, and various infections. Little is known about the rehabilitation needs of this population. This study took advantage of unique access to a specialist access GP practice for people experiencing homelessness and a local inclusion health initiative to explore the five-year period prevalence of these conditions in a population of people experiencing homelessness through electronic case record searches and to identify barriers and facilitators to healthcare provision for this population in the context of an interdisciplinary and multispecialist inclusion health team through semi-structured interviews with staff working in primary and secondary care who interact with this population. The five-year period prevalence of TBI, infections, and mental health disorders was 9.5%, 4%, and 22.8%, respectively. Of those who had suffered a brain injury, only three had accessed rehabilitation services. Themes from thematic analysis of interviews included the impact of psychological trauma, under-recognition of the needs of people experiencing homelessness, resource scarcity, and the need for collaborative and adaptive approaches. The combination of quantitative and qualitative data suggests a potential role for rehabilitation medicine in inclusion health initiatives.

## 1. Introduction

People experiencing homelessness (PEH) suffer from a variety of adverse healthcare outcomes, including (but not limited to) increased cardiovascular disease risk, increased risk of a variety of infectious diseases (tuberculosis/human immune deficiency virus/viral hepatitis C/sexually transmitted infections), psychiatric diseases, and increased health service use more broadly [1,2,3,4,5]. PEH and vulnerably housed people also suffer from increased rates of traumatic brain injury (TBI) and cognitive impairment [4,5,6,7,8,9]. The association between TBI and homelessness is multifaceted, with some evidence suggesting that TBI may be a risk factor for becoming homeless and brain injuries being associated with greater mortality, morbidity, and social exclusion, including incarceration, in PEH [10,11,12,13,14,15]. While there is a significant amount of literature detailing the health and social care needs of this population, there is less focus on identification of the rehabilitation needs of this population, particularly considering the increased prevalence of brain injury within this group.

Rehabilitation following brain injury appears to be an effective intervention. A Cochrane meta-analysis found strong evidence of benefit from “milieu-orientated” community-based rehabilitation programs. In addition, there is some evidence that specialist multidisciplinary rehabilitation programs in both community and in-patient settings appear to be effective in people who have had an acquired brain injury. The review, however, recognizes the heterogeneity in the presentation of brain injury and the need for targeted interventions for specific individuals and specific populations. A further study exploring the cost effectiveness of specialist in-patient rehabilitation following brain injury in the U.K. demonstrated that it is a cost-effective intervention through decreased care costs and improvements in functional independence [16,17].

Despite the recognition of the increased burden of TBI on people experiencing homelessness, there is little consensus on how best to address this unmet need. Research looking into the rehabilitation needs of people experiencing homelessness has demonstrated high levels of physical limitation and cognitive and functional impairments. A scoping review also demonstrated there are existing programs of rehabilitation targeting this population with opportunities to further tailor existing services to better meet the needs of people experiencing homelessness [8,18,19,20]. Policy guidance regarding the health of people experiencing homelessness is clear: Identification of brain injury and management of its sequelae should be priority areas in inclusion health efforts [2,18]. Despite this, a recent review looking at clinical practice guidelines for management of brain injury found most guidelines did not have any special provision for management of brain injury in people who are experiencing homelessness [21].

This study took advantage of the unique access to a local specialist access primary care provider specializing in people experiencing homelessness and vulnerably housed people and a local in-reach inclusion health service, which had a presence in local acute hospitals.

The objectives of the study were to (1) estimate the five-year period prevalence of TBI, in a cohort of vulnerably housed people and people experiencing homelessness, registered to a single specialist access primary care provider and (2) explore the perceived barriers and facilitators to healthcare provision for people experiencing homelessness through interviews with professionals working in a specialist access GP practice for vulnerably housed and homeless individuals, a local brain injury rehabilitation service, and a local inclusion health initiative.

## 2. Materials and Methods

### 2.1. Quantitative

A retrospective case-record analysis was performed on a cohort (*n* = 2753) of individuals registered to a single specialist access primary care provider. This involved performing electronic health record searches to determine the five-year period prevalence of TBI, infectious diseases associated with deprivation (HIV, viral hepatitis, tuberculosis, and sexually transmitted infections), and mental health disorders (including substance misuse disorders). ICD-10 codes (see Appendix A) corresponding to these diagnoses were used to identify relevant cases. Case record analysis was carried out in February 2023 by the e-Health team at the relevant local health board, NHS Lothian (see Acknowledgments Section) using a structured query language program to access data from the local health board electronic health record database (TRAK Care ©). Five years was used as a cutoff for ease of data extraction and feasibility in the context of the limited timescale for the student selected component (SSC) project (10 weeks), alongside the fact that health records within the local health board had only more recently been completely digitized.

### 2.2. Qualitative

Nine semi-structured face-to-face interviews were conducted with four nurses, two medical doctors, two occupational therapists, and one administrative staff member. Two interviews involved multiple participants. Interviewees worked across a local access general practice, a local in-reach inclusion health service, and a local neurorehabilitation service. Interviews were conducted privately, in clinical environments during the initial project window, which was between 16 January and 24 March 2023. Each interview was conducted by two of the authors. All interviews were conducted using an interview guide that was constructed and agreed upon by all authors. Interviews explored the perceived barriers and facilitators to each individual service meeting the needs of their client group.

A purposive and opportunistic approach to sampling was taken. This involved focusing on individuals known to work within local inclusion health and brain injury rehabilitation services, considering the availability of potential interviewees. All interviews were recorded after gaining prior consent from interviewees. They were transcribed using the transcription feature in Microsoft Word^®^. They were subsequently analyzed using a reflexive thematic analysis (as codified initially by Braun and Clarke) [22] with an inductive approach taken to coding and generation of themes. Initial coding was performed independently by four authors. They then met and went through a process of cross comparing initial codes and development of initial themes, which all four authors agreed upon. This was undertaken with a constructivist approach to analysis, with a focus on themes exploring meaning rather than summarizing information. Themes were subsequently developed, re-organized, and reviewed in an iterative process, which involved discussion between all authors. This was an ongoing process for over a year, until submission of the manuscript. The make-up of interviewees (health and social care professionals) and of authors (medical students and physicians) influenced the nature of the interviews, and ultimately, the codes and themes generated reflected this. The final themes reflect “patterns of shared meaning” that are methodologically consistent with our reflexive approach, as further defined by Braun and Clark [23,24].

## 3. Results

### 3.1. Quantitative

Demographic information was collected for the cohort of individuals registered with the access practice (see Table 1). Of the 2753 patients in the cohort, 628 had at least one diagnosed mental health disorder, 262 had at least one episode of TBI, and 111 had a diagnosis of at least one infection associated with deprivation within the last five years (five-year % period prevalence of 22.8%, 9.5%, and 4%, respectively—see Figure 1). Despite the prevalence of TBI, merely three people (0.11%) in the cohort had documented access to brain injury rehabilitation services during this period.

### 3.2. Qualitative Analysis

Four themes arose from thematic analysis of the qualitative interviews. They are further described (in each subsection) and explored below.

#### 3.2.1. The Effect of Psychological Trauma

The effect of psychological trauma and stigmatization was identified in most interviews as a barrier to service delivery for PEH. There was a recognition that previous experiences and treatment of PEH by various services contribute to this:

“Many of our clients, particularly those with addictions, feel quite stigmatized. So, the environment in general can be quite difficult for them and just feeling they’ve got somebody there who’s on their side and advocating for them can make a huge difference.” Inclusion Health Program Manager

“One of the biggest challenges is getting to engage in the first place, you know, just getting them to trust those services because that’s been broken in the past, previous experiences in mainstream GP practices where they’ve stigmatized, and they’ve just turned away for the wee least outburst. Whereas, we have quite a high tolerance level for people’s behavior. I don’t mean we accept really bad behavior; we accept that somebody’s in distress a lot more and we try.” Nurse at Access GP Practice

“Yes, so main obstacles … a lot of mistrust like. A lot of people feel that they have had negative experiences with other GP surgeries, and there have felt a lot of stigma, yeah. So, it’s all about that trust building, isn’t it?” Nurse at Access GP Practice

Training in “trauma-informed care” was seen as a potential enabler in improving outcomes and care in this group of PEH:

“Mental health is a big issue and anxiety, and I would argue that we try to be as trauma informed as we can and have started the whole process and becoming more trauma informed…” Nurse at GP Access Practice

“I’ve done some trauma-informed practice training of my own back.” Doctor working within local In-Reach Inclusion Health Service

“…we try to be as trauma-informed as we can and you know have it, have started the whole process and becoming more trauma-informed.” Doctor working within local In-Reach Inclusion Health Service

#### 3.2.2. Under Recognition of the Needs of PEH

Under recognition of the needs of PEH was another theme that arose from analysis of interviews. This was manifest in two ways. Firstly, there was expression of the perception that health services underestimate the extent of medical comorbidity within this cohort:

“…the average age of death in Edinburgh if you’re homeless is 41 for a woman and 47 for men, 87% had morbidities of the same number as a cohort of the over 80 s. So huge, huge multi-morbidity, very frail, but young cohort, and so all the services available to elderly patients, which are not available [to them].” Doctor working within local Inclusion In-Reach Health Service

Secondly, in the context of recognition of the sequelae of brain injury specifically, interviewees from various services noted a perception of underestimation of the prevalence of brain injury in this population. This lack of recognition may account for the relatively small numbers of PEH in the cohort who had access to rehabilitation services and suggests a potential need for more comprehensive screening for the rehabilitation needs of PEH:

“I think brain injury in general is vastly under-recognized, if you compare it to something like stroke or maybe cancer services, you know you’re talking about equally life changing illnesses, and [they] also affect… usually affecting younger people. So, they are going to live with this for a longer period of time, so, no, brain injury is vastly under-recognized and under-resourced, I would say…” Specialist Brain Injury Occupational Therapist

“…our inclusion health huddle on a Wednesday, we have hepatitis in reach, nurse drug liaison who are really important third sector. I mean, we haven’t really thought about neurorehabilitation. But now I am…” Doctor working within local In-Reach Inclusion Health Service

“…people will often. Maybe not have been really assessed for a brain injury because if they have presented previously following an accident and self-discharged any assessment’s quite difficult…” Inclusion Health Program Manager

#### 3.2.3. Resource Scarcity

The issue of resources is a pertinent one across the health service and cited by staff across various services as a barrier to care. This was manifest in the context of a recognition of limited resources for services providing care for people experiencing homelessness and other marginalized groups, including people who had suffered brain injuries. There was a sense that a lack of adequate resources was potentially leading to less positive outcomes for clients than might otherwise be achieved:

“I think it’s true of all aspects of the NHS, but resources, you know, not having enough people to be able to see patients and have, you know, particularly people with quite significant cognitive impairment, you would want to be able to do repetition to try and support some need to improve and cope and build strategies, but if you don’t, you’re not able to do that repetition because you don’t have adequate staffing to do that either as an inpatient or an outpatient. I think that leads to like skewed outcomes for patients.” Specialist Brain Injury Occupational Therapist

“Oh yes, we are always up to capacity. The difficulty is because we’re a small team and we’ve had. In three years, we have had about 13, 14 hundred referrals, so the difficulty we have is that we can offer that long-term support to everybody.” Inclusion Health Program Manager

#### 3.2.4. Collaborative and Adaptive Approaches

Collaboration and integration were the major enabling factors identified in our analysis. This was the most consistent theme across services and throughout the interviews. Collaboration was a clear facilitator both within the context of viewing the care of patients as an endeavor requiring partnership and as an important factor for the various services involved with the population of PEH:

“…yes, so there’s a number of different things, one of the challenges is they’ve got a lot of other pressures going on in their lives as well, so if you’re looking at health side of things we are now integrated with social work, health, and housing, which supposedly makes access a bit better.” Primary Care Doctor

“Oh no, we absolutely link up. We linked up with hundreds of services. So, all the health services GP practices, community support organizations, addiction services. Just too many to mention. We link up, that is the key, I mean, we do not have the capacity to provide large packages of support on an ongoing basis. So, linking people up with a variety of community support that is tailored to their needs is much more sustainable for them in the future.” Inclusion Health Program Manager

The importance of centering patients and offering a degree of adaptability in care to ensure the needs of the most vulnerable are met was emphasized by staff in multiple interviews. The importance of adopting a pro-active and flexible model to healthcare delivery was cited as a strength within the collaborative inclusion health model, which involved integration between various teams working across in-patient, community healthcare, and social services:

“So, what healthcare can we deliver in an alternative setting in that situation?… So, we have to get rid of that gold-standard treatment… those guidelines are written in without the patient really in mind. And if that’s not tolerable, then what’s the next best thing?” Doctor working within local In-Reach Inclusion Health Service

“There’s a lot of people who come incredibly sporadically who are most needy. Given we have an opportunistic service they might see housing, social work, and health and the nurse and a welfare advisor all in one morning.” Nurse GP Access Practice

## 4. Discussion

### 4.1. Contextualising Quantitative Results

The prevalence of head injury, mental health disorders, and infections was consistent with previous studies [1,3,4]. The prevalence of brain injury in this cohort was also consistent with previous studies exploring TBI in people experiencing homelessness [2,6,7,10]. Previous literature has largely relied on methodology based on self-reports of head injury that may not necessarily be corroborated and easily verifiable. This is often further complicated if injuries occur in the context of intoxication, as this is a risk factor for head injury. Given the above limitations, a pragmatic approach was taken to identify cases of TBI through searching for ICD 10 clinical codes associated with a traumatic head injury (see Table A1). It is possible that hospital episodes that included head injury in the context of polytrauma or coincidence of head injury with other diagnoses may not have been coded to reflect this. Additionally, the methodology used only captures hospital-treated brain injury, which may also contribute to underestimation due to individuals not presenting to secondary care services following head injury. The low number of individuals in the cohort accessing rehabilitation services following a TBI was particularly noteworthy. While the benefits of rehabilitation following brain injury are becoming increasingly clear, there are few data on the demographic characteristics of patients accessing rehabilitation services. Several studies have demonstrated significant functional impairment in people experiencing homelessness [8,25,26,27], and some studies have demonstrated some benefit from a rehabilitative approach focused on addressing functional impairments both due to brain injury and other causes.

### 4.2. Contextualising Qualitative Results

The lack of identification of trauma and appropriate response by healthcare staff limits the potential therapeutic relationships that necessarily underpin health and social care. The impact of trauma and stigmatization has been explored in other qualitative studies exploring healthcare for PEH and more broadly [28,29,30,31,32]. Wider training in trauma-informed care represents a potential means to improve this. This approach is well recognized and practiced in various contexts within health and social care [33]. It involves a five-stage approach: trauma awareness; safety and trustworthiness; choice and collaboration; building skills and resilience; and recognition of wider cultural, historical, and gender issues (intersectionality) [34]. This approach would seem to be aligned to the holistic ICF framework adopted by rehabilitation professionals in their approach to the assessment and management of disability, which takes into account disease-related, functional impairment, and wider contextual factors such as environment and personal factors [35].

As stated by multiple interviewees, the population of PEH has an extent of multimorbidity usually associated with much older populations; however, unlike the older population, they do not tend to have the same recognition of this among health care professionals. The impact of frailty secondary to multimorbidity has been explored and characterized in other studies [36,37]. Any attempts to improve the health of PEH need to take this into account and will need to be sufficiently holistic to address this. A rehabilitation-focused approach with a focus on disability in a more holistic sense rather than a focus on individual diagnoses may be of value in this context.

The issue of resource scarcity is pertinent across all areas of healthcare and affects all who interact with services. A study looking at health resource allocation and its impact on health inequalities in the U.K. suggested that lower resource allocation disproportionately affects more marginalized groups [38]. Given the burden associated with increased health service use associated with homelessness and evidence of the benefit of multidisciplinary rehabilitation following brain injury, integration of rehabilitation into community-based health programs for PEH might have the potential to deliver cost savings [2,16,17].

Patient-centered care is a core theme in many policy documents around organization of healthcare services, including rehabilitation services [39,40,41]. However, the extent to which patient-centered care is implemented in healthcare systems, including rehabilitation, is unclear [42]. Rehabilitation services collaboration within such efforts may lead to further appropriate input with PEH in the context of rehabilitation following TBI, particularly given the relative paucity of rehabilitation input identified from the quantitative aspect of the study.

## 5. Conclusions

This study is unique in several ways: Firstly, it is novel in its approach of identification of TBI through clinical coding rather than reliance on self-reporting, and as far as the authors are aware, it is also the first to explore access to rehabilitation post brain injury in people experiencing homelessness. Our mixed-methodology approach allowed us to assess prevalence of TBI and other conditions while simultaneously gaining insights into the perceptions of the relationship between brain injury, other conditions, and homelessness. We believe that this study suggests a potential role for integration of brain injury rehabilitation services into existing pathways of inclusion health as a means of mitigating the significant level of morbidity within PEH and other marginalized groups at increased risk of brain injury. Rehabilitation medicine specialists focus on all aspects of disability (impairment, function, participation, personal, and environmental factors), and expertise makes the specialty particularly well suited to support inclusion health efforts in a holistic fashion, traversing the biopsychosocial paradigm [43,44]. Such models coincide with the (sub)theme-identified enablers to healthcare provision for people experiencing homelessness. The presence of embedded local inclusion health teams in primary care with a significant at-risk population provides a significant opportunity for further work to screen for TBI and its resultant impairments and design rehabilitation interventions with the aim of mitigating their effects. Further research combining qualitative and qualitative methodology should involve existing stakeholders and most importantly service users to explore how this might best be achieved. Such work might best begin with soliciting the views of PEH themselves on how service design can better suit their needs and building on the work that has already been done in this area [45,46].

## Figures and Tables

**Figure 1 ijerph-21-00769-f001:**
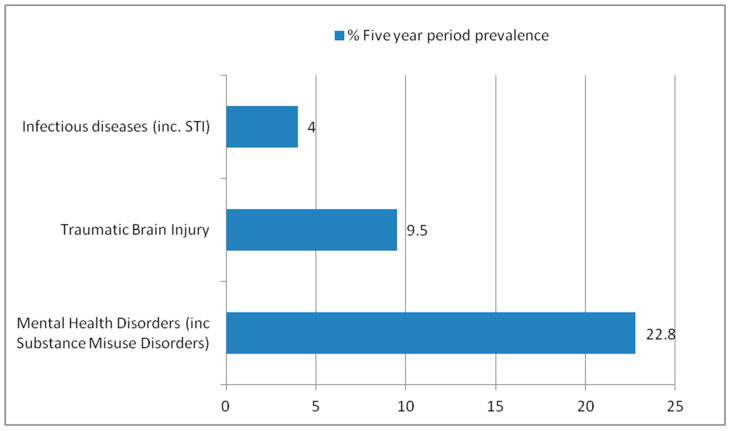
Five-year period prevalence of Infectious Diseases, Traumatic Brain Injury, and Mental Health disorders among cohort (*n* = 2753).

**Table 1 ijerph-21-00769-t001:** Demographic Information of Access GP Practice Patient Cohort (*n* = 2753).

**Ratio of Female/Male**	1:3.1 *
**Average Age**	47
**Ethnicity ****	African, African Scottish, or African British—11 Any mixed or multiple ethnic group—9 Any other white ethnic group—17 Arab—1 Australasia (Australia, New Zealand)—2 Bangladeshi, Bangladeshi Scottish, or Bangladeshi British—1 Black, Black Scottish, or Black British—5 Chinese—1 Chinese, Chinese Scottish, or Chinese British—4 E Europe exc. Poland (e.g., Balkans, Russia)—36 Indian, Indian Scottish, or Indian British—1 N Europe (e.g., Denmark, Norway, Sweden)—2 Other Asian—4 Other Black—5 Pakistani, Pakistani Scottish, or Pakistani British—2 S Europe (e.g., Cyprus, Greece, Italy, Spain, Turkey)—16 W Europe (e.g., France, Germany, Netherlands)—6 White British—238 White English—30 White Irish—8 White Northern Irish—2 White Scottish—532 White Welsh—2

*: Information for sex available for 2751 people (unknown for remainder of cohort); **: Information for ethnicity available for 935 (undocumented or information not supplied for remainder of cohort)

## Data Availability

All original audio recordings from interviews were destroyed, and transcripts are only available via drives accessible from authorized University of Edinburgh account holders. Data sharing is not applicable to this article.

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
