# Peer review of "Exploring the Role of Rehabilitation Medicine within an Inclusion Health Context: Examining a Population at Risk from Homelessness and Brain Injury in Edinburgh"

_ijerph, 2024, doi:10.3390/ijerph21060769_

Round 1
Reviewer 1 Report
Comments and Suggestions for Authors
Thank you for inviting me to review this paper. This paper reports on an important and timely study that has the potential to add to the literature on this topic. The mixed methods approach is a clear strength of this study, as it highlights the prevalence of TBI in PEH as well as considering - with input from professionals - how this might be best addressed. However, in my review, the qualitative aspects of the paper in particular require further attention and some re-structuring is required.
Introduction
Line 44 - It would be beneficial to provide slightly more detail about what is known to be effective in terms of rehabilitation interventions - this seems quite a broad statement at present.
Line 57 - This should be a new paragraph, as it was unclear to me that you had begun speaking about your own study.
Materials and Method - Section 2.2.
You use the term questionnaire to describe the material used during your interviews (Line 86). Given your semi-structured approach, the term 'interview guide' would be more appropriate here; an interview guide differs quite considerably from a questionnaire.
Could you comment on the setting in which the interviews were conducted, and also when the data collection took place?
Much more detail about the coding/analysis process is also required here - for example, what strategies were utilised to allow you to move from the process of initial coding to the generation of themes? There is a mention of four authors being involved in this process - did each complete some of the coding? At what point was the coding of different authors compared/integrated? How - as a team - were themes developed? How were differences in opinion managed? It would also be useful to include whether any software was used to aid this process, or whether this was completed manually.
The authors may find it beneficial to consider engaging with a more recent version of thematic analysis (for example, 2022) as Braun and Clarke have significantly developed their guidance sine 2006.
Some further comment regarding the ethical considerations in place for this study would also be helpful. For example, how was data stored?
Results/Discussion
While the data presented is of good quality and the commentary alongside it is well considered, I feel that these parts of the paper require some re-structuring. The main concern I have is that I do not think the qualitative data should not be presented in the Discussion - the data (and your analysis of it) are empirical findings and should therefore be placed in the Results section. I would therefore suggest that within the qualitative part of your Results section, you use each of your themes as a subheading, and then organise your empirical evidence under these subheadings, explaining and evidencing each in turn.
Doing this would mean that the Discussion could be used to more effectively discuss the implications of the study findings in terms of rehabilitation interventions - this seems to be an extremely important element of this particular study, and at present is only rather briefly discussed in the Conclusion.
Comments on the Quality of English LanguageClear and well written - just a few minor typos.
Author Response
Thank you for your comments and for taking the time to review our submission. I will address your comments/suggestions below and provide an updated manuscript in due course.
- With respect to the statement on line 44, I appreciate your point about the statement being rather generalised. The references related to this are based on cocharne review suggesting evidence of functional improvement (heterogenous outcome measures) following multidisciplinary inpatient specialist rehabilitation after acquired brain injury and UK data suggesting that specialist inpatient multidisciplinary rehabilitation is a cost effective intervention due to reduction in care costs and functional indepndence gains. We can make this point clearer and perhaps provide more evidence from elsewhere to bolster this claim.
- I am happy to make line 57 a new paragraph for clarity.
- I take on board your clarification on the distinction between a questionnaire and interview guide, and am happy to change the wording to reflect this
- I can elaborate on the setting for interviews, all of which were conducted in private rooms in clinical environments. The window for data collection was during the timeperiod for the SSC placement which was 10 weeks from January to March 2023.
- In terms of the analysis of qualitative data, initial coding was done independently by 4 different authors. We then came together to amalgamate coding and work through this in an initial meeting. At this point we identified that codes could largely be separated into enablers and barriers. We then had a further meeting where we developed initial themes. These were then discussed and refined, and presented to the rest of the authours multpile times both prior to and during draft manuscripts, until we eventually all agreed on the final themes. I can elobrate on this process further in the amended manuscript.
- I am aware of subsequent revisions, and development of the guidance on thematic analysis, and we are happy to reference a more recent overview as suggested
- In terms of data storage, quantitative data was stored on NHS Lothian hardrives only accessible on NHS Lothian computers in accordance with health board information governance policy. Transcripts of interviews were stored on University drives, again only accessible on university accounts/computers. Ethical approval to use both qualitative and quantitative data was saught and granted from both the university of Edinburgh and health board. Original recordings of all interviews were deleted
- With regards to the results and discussion, we accept your point and are happy to reformat to include qualitative analysis within the results section and eloborate further on the main discussion points
Many thanks again for taking the time to review our submission and for your insightful comments.
Reviewer 2 Report
Comments and Suggestions for Authors
I found this paper exceptionally clean, clear and easy to read making this review an easy task. Traumatic brain injury sequelae and access to treatment for PEH is an interesting and under researched topic. The resources for TBI are often limited for all and as you note, very difficult for PEH to access if available.
I think the mixed methods approach you used worked well in this study providing some quantitative context supported by clinical observations. Your sampling technique seemed to bring you those clinicians with significant experience with PEH thus providing a rich and relevant data source. For me, collaboration, training and trauma informed care are key to working with PEH on any level. The need to be present to the needs of PEH whatever and wherever they are and follow-up- doing what you say you will do, go a long way to forming trusting therapeutic relationships that too are key to working with this group of people.
I have several questions/comments:
1. You used five years as your cutoff for quantitative data. You state this was for feasibility and data extraction issues. Can you elaborate a little more on this? Another sentence or two would help me contextualize you data a little more.
2. Lines 126/7 state, "This is often further complicated if injuries occur in the context of intoxication, as is often the case in this population." I would ask for a reference either to your data or a paper here, otherwise this language seems a little presumptive and stigmatizing to me.
3. I think that lines 199-200 might benefit from a note that PEH often fall through the cracks here as everywhere because they may have nowhere to live, are hard to contact, need time to make connections, are often reluctant to access health services and can take a long time to gain trust leaves them at the bottom for the few services that do exist.
4. I think perhaps a little more detail on how service users might be meaningfully included in future research in this area and inclusion medicine in general could add to the significance of the findings in this paper. Inclusion is vastly different from consultation and participation, often just 'lip service' is paid in medical services as I imagine you already know.
Thank you for asking me to review this paper. Much good luck to you!
Author Response
Thank you for the taking the time to review our paper. Please see below our response to the queries raised.
- 5 years was used as a cut off primarily on the advice of the data analytics team at the health board. This was partly due to the limited amount of time we had during the SSC project data collection window 10 weeks, and also due to the fact that health records locally were not fully digitised until fairly recently meaning older data would have been less reliable.
- I accept your point about the language used having the potential to casue further stigmitastion (precisely of the kind we wish to avoid attaching to PEH). Perhaps we can remove this sentence or rephrase it in such a way as to recognise that alcohol is a risk factor for head injury accross all populations.
- We accept your suggestion and will reference this
- We have elaborated on this in some more detail and suggested ways in which future research can meaningfully engage PEH, including references from previous work in this area
Round 2
Reviewer 1 Report
Comments and Suggestions for Authors
Previous comments have been thoughtfully addressed. No further comments to provide - thank you for the opportunity to be involved in the review of this paper.